# HDAC Inhibitors in Acute Myeloid Leukemia

**DOI:** 10.3390/cancers11111794

**Published:** 2019-11-14

**Authors:** Edurne San José-Enériz, Naroa Gimenez-Camino, Xabier Agirre, Felipe Prosper

**Affiliations:** 1Área de Hemato-Oncología, Centro de Investigación Médica Aplicada, Instituto de Investigación Sanitaria de Navarra (IDISNA), Universidad de Navarra, 31008 Pamplona, Spain; esanjose@alumni.unav.es (E.S.J.-E.); ngimenez@alumni.unav.es (N.G.-C.); 2Centro de Investigación Biomédica en Red de Cáncer (CIBERONC), 28029 Madrid, Spain; 3Departamento de Hematología, Clínica Universidad de Navarra, Universidad de Navarra, 31008 Pamplona, Spain

**Keywords:** acute myeloid leukemia, epigenetic, histone deacetylases, histone deacetylases inhibitors

## Abstract

Acute myeloid leukemia (AML) is a hematological malignancy characterized by uncontrolled proliferation, differentiation arrest, and accumulation of immature myeloid progenitors. Although clinical advances in AML have been made, especially in young patients, long-term disease-free survival remains poor, making this disease an unmet therapeutic challenge. Epigenetic alterations and mutations in epigenetic regulators contribute to the pathogenesis of AML, supporting the rationale for the use of epigenetic drugs in patients with AML. While hypomethylating agents have already been approved in AML, the use of other epigenetic inhibitors, such as histone deacetylases (HDAC) inhibitors (HDACi), is under clinical development. HDACi such as Panobinostat, Vorinostat, and Tricostatin A have been shown to promote cell death, autophagy, apoptosis, or growth arrest in preclinical AML models, yet these inhibitors do not seem to be effective as monotherapies, but rather in combination with other drugs. In this review, we discuss the rationale for the use of different HDACi in patients with AML, the results of preclinical studies, and the results obtained in clinical trials. Although so far the results with HDACi in clinical trials in AML have been modest, there are some encouraging data from treatment with the HDACi Pracinostat in combination with DNA demethylating agents.

## 1. Introduction

Acute myeloid leukemia (AML) is a heterogeneous clonal disorder characterized by the uncontrolled proliferation, differentiation arrest, and accumulation of immature myeloid progenitors in peripheral blood and bone marrow [1,2,3,4,5,6]. AML is the most common form of acute leukemia in adults, accounting for 80% of all patients diagnosed with acute leukemia. AML is a disease of the elderly, with a median age at diagnosis of 70 years. With increasing incidence in recent years, the overall incidence of this disease is 3.4 cases per 100,000 people, with 1.2 cases per 100,000 population at age 30, and more than 20 cases per 100,000 population at age 80 years [3,4,5,6,7,8]. Despite significant advances in understanding the molecular pathogenesis of the disease, AML treatment has remained unchanged for the past four decades and is still largely based on standard chemotherapy and the use of allogeneic hematopoietic stem cell transplantation. However, new drugs have been recently approved by the United States Food and Drug Administration (FDA) for AML, such as Gemtuzumab ozogamicin, enasidenib, midostaurin, glasdegib, venetoclax, ivosidenib, and gliteritinib [9]. Although disease-free survival and overall survival have improved in recent years, improvement has been mostly observed in younger patients. In fact, approximately 40% of patients younger than 60 years of age may obtain long-term disease-free survival with current therapy. However, in older patients, survival is significantly worse, and in fact there has been very limited improvement [3,4,5,6,8,10]. Unfortunately, older patients show a high rate of relapse with a notoriously poor outcome. Considering that AML occurs more commonly in older patients, this represents a huge challenge for AML drug therapy [3].

Historically, AML has been considered a genetic disease, but a number of recent studies have demonstrated that epigenetic changes participate in the pathogenesis of AML [11,12,13]. DNA methylation and histone modifications are the most extensively described epigenetic modifications associated with regulation of chromatin structure and gene expression. These modifications are catalyzed by epigenetic regulatory enzymes, such as DNA methyltransferases (DNMTs), histone methyltransferases (HMTs), histone acetyltransferases (HATs), and histone deacetylases (HDACs) [14,15,16]. In fact, most of these enzymes have been shown to be deregulated or mutated in several cancers and specifically in AML [15,16,17,18]. This fact, together with the evidence that epigenetic modifications are pharmacologically reversible, has placed epigenetic drugs at the center of both clinical and preclinical research in AML.

Since the role of DNA methylation and the effect of hypomethylating agents in AML has been recently reviewed [19,20], we will focus on the rationale for the use of HDAC inhibitors (HDACi) for AML treatment. Additionally, we will discuss the novel evidence regarding the synergistic effects, in preclinical and clinical studies in AML, of HDACi with DNA hypomethylating agents and other inhibitors.

## 2. Histone Acetylation

Nucleosomes are the fundamental units of chromatin and are composed of an octamer with four-core histones (histone 2A (H2A), histone 2B (H2B), histone 3 (H3), and histone 4 (H4)), with 146 base pairs of DNA wrapped around these histones. Each histone is composed of a structural domain and an unstructured amino terminal tail of 25–40 residues. Importantly, these histone tails can be altered by different post-translational modifications that can determine the interactions between the histone and other proteins [21], playing a key role in the transition between active and inactive chromatin states [22].

Among these modifications, one of the best characterized is the (de)acetylation of the histone tail. In this process, the lysine residues present mainly in the N-terminal region of histone tails are acetylated or deacetylated, leading to gene regulation. Acetylation leads to the neutralization of positive charge in lysine, and therefore, to weakening of the electrostatic interaction between histone and negatively charged DNA. Thus, histone acetylation opens the chromatin structure, allowing for the binding of transcription factors, leading to gene expression. Meanwhile, histone deacetylation confers a close chromatin structure and inhibits or decreases gene transcription. Regulation by histone (de)acetylation is involved in important processes, such as replication, chromatin packing, DNA repair, nucleosome-non-histone proteins interactions, cell apoptosis, senescence, differentiation, angiogenesis, or immunogenicity [21,23]. Chromatin immunoprecipitation followed by sequencing (ChIP-Seq) analyses have shown that histone acetylation is distributed in genomic regions defined as promoter or enhancer regions, and also throughout the transcribed regions of active genes [16,17,24,25]. It is important to emphasize that several proteins are also able to recognize and bind acetylated histone tails, such as bromodomain proteins or proteins with plant homeodomain (PHD) finger proteins [26].

Histone (de)acetylation is a highly dynamic process regulated by the epigenetic enzymes HATs and HDACs. These two large families of enzymes have opposite actions: HATs catalyze the transfer of an acetyl group from acetyl coenzyme A to the lysine residues of the histones [27,28] and are mainly divided into three major subfamilies, the GCN5-related N-acetyltransferase (GNAT), MYST, and CREB binding protein (CBP)/p300. HDACs instead catalyze the removal of acetyl functional groups from the lysine residues of the histones (Figure 1) [27,28,29], and 18 HDAC enzymes have been identified in mammals [23,30].

### 2.1. HDAC Classes

HDACs are divided into four family classes, with each class showing a different subcellular location and specificity. Class I HDACs includes HDAC1, HDAC2, HDAC3, and HDAC8, which are located mostly in the nucleus. Class II is in turn divided into two groups: class IIa, including HDAC4, HDAC5, HDAC7, and HDAC9; and class IIb, with HDAC6 and HDAC10. All of them are capable of shuttling between the nucleus and the cytoplasm, depending on different signals [18,31]. Class III includes the sirtuin (SIRT)1-7 sirtuin protein family. Finally, class IV only has one member, HDAC11, which is located both in the nucleus and in the cytoplasm. Class I, II, and IV HDACs are all Zn^2+^-dependent enzymes. However, class III HDACs require NAD^+^ instead of Zn^2+^ as a cofactor and they do not have protein sequence homology with the other HDACs classes (Table 1) [18,31].

### 2.2. HDACs: More Than Histone Deacetylases

It is well reported that global histone (de)acetylation levels are aberrantly altered in cancer, but HDACs are much more than histone deacetylases. Acetylation is a common post-translational modification of non-histone proteins that are also acetylated and deacetylated. Among these non-histone proteins, there are oncogenes, tumor-suppressor genes, transcription factors, chaperones, and cell signaling molecules, such as MYC, P53, or PTEN, which result in changes in protein stability, protein–protein interactions, and protein–DNA interactions [16,17,32]. Both histone and non-histone (de)acetylations are important regulators of different biological processes of the cells. In a general sense, HDACs should be more correctly called lysine deacetylases (KDACs).

One of the best characterized non-histone substrates among the HDACs is the tumor suppressor P53, which induces anti-proliferative effects in the cells, including growth arrest, apoptosis, and cell senescence, in response to various types of stress [33,34]. P53 acetylation, by p300/CBP or by the p300/CBP-associated factor (PCAF), increases its DNA-binding ability and consequently the transcriptional activation of its target genes, finally inducing the cell cycle arrest or apoptosis, depending on the cell type and nature of the cellular stress. However, deacetylation of P53 can be mediated by two different HDACs: HDAC1 and SIRT1. The HDAC1 complex specifically interacts with P53, inducing deacetylation and leading to its degradation. SIRT1 interacts with P53, inducing deacetylation of multiple lysine residues and suppressing the P53-dependent growth arrest and apoptosis, resulting in an increased risk of tumorigenesis [34].

HDACs may also play a role in the regulation of the immune system by targeting the transcriptional regulator STAT3 [34,35]. STAT3 is implicated in the regulation of growth and apoptosis of immune cells, playing a role in a variety of autoimmune diseases. This transcriptional regulator can be acetylated on a single lysine residue at position 685 by histone acetyltransferase p300, and deacetylated by class I HDACs. Interestingly, acetylation of STAT3 is critical for its stabilization, and thus for cytokine-stimulated DNA binding and transcriptional regulation [34,36].

The cytoskeletal protein α-tubulin is another non-histone protein that is regulated by dynamic acetylation and deacetylation. Reversible acetylation of α-tubulin has been implicated in the regulation of microtubule stability. The acetyltranferase responsible for this acetylation remains unknown, whereas HDAC6 and SIRT2 were found to deacetylate lysine at position 40 of α-tubulin in vitro and in vivo. HDAC6 over-expression leads to hypoacetylation of α-tubulin, resulting in an increase in cell motility. Furthermore, the epithelial mesenchymal transition mediated by the transforming growth factor beta 1 is accompanied by the HDAC6-dependent deacetylation of α-tubulin [34,37].

### 2.3. Implication of HDACs in Cancer

HDACs are important proteins that are directly implicated in the epigenetic regulation of gene expression and the control of cellular activities by reversing the histone acetylation status [38]. In essence, changes in chromatin structure due to histone (de)acetylation might result in decreased or increased gene transcription, altering gene expression levels. Indeed, gene expression determined by HDAC inhibition is not always increased, although the chromatin structure is loosened [21,39]. Therefore, it is not surprising that the deregulation of HDACs is related to the development of several diseases, particularly cancer. Several types of human tumors, including gastric, colorectal, liver, breast, lung cancers, and hematological malignancies, show aberrant HDAC expression, in many cases associated with advanced disease and poor prognosis in cancer [40,41,42,43,44,45,46]. Nevertheless, the amount of evidence that demonstrates the oncogenic role of HDAC overexpression is limited. A clear example is the overexpression of *HDAC1* in prostate tumor cells, inducing cell proliferation and de-differentiation [47]. Meanwhile, it is well known that knockdown of HDACs lead to cell cycle arrest, decrease of proliferation, and induction of apoptosis or differentiation, among other anti-tumor effects [42].

In addition, somatic mutations of *HDAC* genes are not common events in cancer and their role in tumor development has not been studied in detail. Specifically, mutations of *HDAC1* have been found in human liposarcomas, *HDAC2* mutations have been found in epithelial cancers and colorectal cancer, *HDAC4* mutations have been found in breast cancer, and mutations in *HDAC9* have been found in prostate cancers [31,48,49,50,51]. These mutations may be related to the development and progression of tumors, although further investigation will be required to elucidate the real implication of these genetic alterations in the development or progression of human tumors.

In the case of AML, mutations in *HDACs* genes have not been detected, but interestingly, it has been described how these HDAC proteins are aberrantly recruited to specific gene promoters by abnormal oncogenic fusion proteins that occur in this disease, such as PML-RARα, PLZF-RARα, or AML1-ETO, mediating aberrant gene silencing contributing to leukemogenesis [52]. For instance, AML1-ETO chimeric fusion protein, typical of AML patients with the translocation t(8;21)(q22;q22), recruits HDAC1, HDAC2, and HDAC3, silencing AML1 target genes, and therefore leading to differentiation arrest and transformation [53,54,55]. In addition, the fusion proteins PML-RARα and PLZF-RARα recruit both HDACs and DNA methyltransferases (DNMTs), driving repression of RARα target genes [56,57,58]. Additionally, an interaction between HDACs and non-chimeric fusion proteins, such as BCL6, whose activity is controlled by acetylation, has been described in AML [59].

## 3. Histone Deacetylase Inhibitors (HDACi): Mechanism of Action and Role in AML

Histone deacetylase inhibitors are a family of natural and synthetic compounds that inhibit the functional activity of HDACs, altering the regulation of histone and non-histone proteins [23,60]. HDACi activities result in an increase in the acetylated levels of histones, in turn promoting the re-expression of different silenced genes in each cell type [61]. Although the exact mechanism of action of HDACi is still unclear, these compounds play important roles in epigenetic or non-epigenetic regulation in the cells, inducing cell death, apoptosis, differentiation, and cell cycle arrest in cancer cells [23,60].

Based on their chemical structures and enzymatic activities, HDACi can be classified most commonly into five groups: hydroximates, benzamides, cyclic tetrapeptides, aliphatic acids, and electrophilic ketones. HDACi may act specifically against one or two types of HDACs (HDAC isoform selective inhibitors) or against all types of HDACs (pan-inhibitors) (Table 2). Zinc-dependent HDACi are characterized by a structure divided into three domains: (1) a cap group or a surface recognition unit, (2) a zinc binding domain (ZBD), and (3) a linker domain that combines the above two parts together. The cap and linker domains contribute to ligand–receptor interactions and affect the selectivity of HDACi, whereas ZBD binds to the zinc ion, inhibiting HDACi activity [62,63].

HDACi has emerged as a promising therapeutic strategy for cancer therapy. Furthermore, these inhibitors have shown ability to induce differentiation, cell cycle arrest, and apoptosis in AML, leading to a good alternative for treatment, especially for those AML patients not suitable for intensive chemotherapy [23,60]. Despite the promising preclinical results of HDACi, these HDACi do not seem to be clinically effective as monotherapies in AML. However, combination strategies with a variety of anticancer drugs are being tested in clinical trials, showing significant anti-leukemic activity in hematological diseases as they enhance the action of some standard-of-care anti-AML treatments [99,145,146].

### 3.1. Hydroximates

Hydroximates were the first HDACi to be discovered, and are thus the most extensively studied, showing strong activities and simple structures [147,148]. These HDACi are unselective inhibitors that target HDAC classes I and II. In these HDACi, the hydroxamic acid binds directly to zinc in the active site pockets of HDACs, mainly through a sulfhydryl group [148].

#### 3.1.1. Trichostatin A

Trichostatin A (TSA) was the first described hydroxamate-based HDACi. It is an antifungal antibiotic isolated from *Streptomyces hygroscopicus* with cytostatic and differentiating properties in mammalian cell culture [149]. Apparently, TSA promotes the expression of apoptosis-related genes, leading to reduced survival of cancer cells, thus slowing the progression of cancer [150,151]. TSA has also been described to induce cell differentiation by inhibition of HDACs in different tumors [152,153,154]. In vitro experiments carried out with AML cell lines showed that TSA decreased the main pathway for DNA repair, namely non-homologous end-joining (NHEJ). Its mechanism of action was the acetylation of repair factors and trapping of PARP1 at DNA double-strand brakes in chromatin, inducing leukemic toxicity and suggesting a possible synergistic effect of TSA with an inhibitor of PARP1 [155]. Other studies showed promising results with TSA in combination with Chaetocin, an inhibitor of the histone 3 lysine 9 methyltransferase SUV39H1, inducing apoptosis of AML cell lines and primary patient samples [64]. Moreover, the triple combination of TSA, the demethylating agent 5-aza-2′-deoxcytidine (Decitabine, 5-AZA-CdR), and the EZH2 inhibitor DZNep induced a significant synergistic anti-leukemic effect in AML cells by re-expressing several tumor suppressor genes [65]. However, this HDACi was only used in the laboratory because of its high toxicity.

#### 3.1.2. Vorinostat

Vorinostat (SAHA) was the first marketed HDACi which promotes protein acetylation, modulates gene expression, induces differentiation, growth arrest, and apoptosis of tumor cells [156,157,158]. By binding to the active site of HDACs, this drug inhibits class I and class II HDAC enzymes, with predominant effects on class I HDACs, and has shown promising clinical activity against different hematological tumors [156,157]. Vorinostat rapidly moved through preclinical and clinical studies, and after several studies was approved by the United States Food and Drug Administration (FDA) for the treatment of cutaneous T-cell lymphoma (CTCL) [40,147,159,160]. Regarding AML, Vorinostat has demonstrated in vitro activity against AML cell lines, inducing apoptosis and inhibiting cell growth, and increasing differentiation induced by retinoic acid (ATRA) of acute promyelocytic leukemia (APL) cells [66,67,68]. These effects seem to be mediated by the induction of double-strand breaks and oxidative DNA damage by Vorinostat [161]. This HDACi also improved survival of mice with APL [67]. In spite of these preclinical promising results, a phase 2 study of Vorinostat as a monotherapy showed minimal activity in patients with relapsed AML and in selected untreated patients with high-risk AML, suggesting that future studies should focus on combinations with other drugs [162].

Numerous studies have demonstrated the synergistic effect in AML of Vorinostat in combination with other compounds, such as the aurora kinase inhibitor MK-0457 [69], the proteosome inhibitor NPI-0052 [70], cytosine arabinoside (also known as Cytarabine), Etoposide [71], the BH3-mimetic GX15-070 [72], the Wee1 inhibitor AZD1775 [73], or the FLT3 inhibitor BPR1J-340 [74], among others. Interestingly, several novel hybrid anticancer drugs with activity against AML cells have been developed, such as the SAHA–Bendamustine hybrid NL-101 [163] and the piperlongumine–SAHA hybrid inhibitor [75]. NL-101, which was developed by replacing the side chain of the alkylating agent Bendamustine with the hydroxamic acid of Vorinostat, presenting properties related to HDAC inhibition, DNA damage, and induced apoptosis of leukemic cells. The combination of the SAHA–Bendamustine hybrid inhibitor with the PARP inhibitor BMN673 had a strong synergistic effect in AML, inducing apoptosis and arresting cell cycle in vitro and prolonging mouse survival in vivo [164]. Once more, the mechanism of action of this drug is the induction of DNA damage. Several clinical trials have been carried out with Vorinostat in combination with several compounds, including Decitabine [76,77], Idarubicin [78], Idarubicin plus Cytarabine [79], Alvocidib [80], Gemtuzumab ozogamicin plus Azacitidine (AZA) [81,82], or Sorafenib plus Bortezomib [83]. The phase II trial of Vorinostat with chemotherapy agents Idarubicin and Cytarabine showed that when combined in this way, this epigenetic drug is safe and active in AML treatment. Indeed, this combination has been associated with high induction response rates without any changes in toxicity compare with Idarubicin–Cytarabine regimen [79]. A phase I/II study of Gemtuzumab ozogamicin in combination with Vorinostat and AZA showed that this triple combination has activity in older patients with refractory or relapsed AML [81]. However, when combining Vorinostat with AZA, overall survival of AML patients did not increase in comparison with those that were treated only with AZA [165]. Vorinostat has also been examined in combination with decitabine, administered either sequentially or concurrently in older patients with untreated AML or patients with relapsed or refractory AML, with both schedules being safe and tolerable. In spite of the number of patients evaluated being small, this study showed a better response in the concurrent schedule (response rate 46%) than the sequential treatment (response rate 14%) [76,77]. Another phase I clinical trial with Vorinostat in combination with Decitabine and Cytarabine in relapsed or refractory AML patients showed that this combination was generally well tolerated, with an overall response rate of 35% [166]. Recently, a phase I trial showed clinical response in poor-risk AML patients with the combination of sorafenib (FLT3/RAF inhibitor) and Vorinostat. Moreover, the clinical response was greater with the addition of Bortezomib to the previous combination (3% complete response (CR), 16% complete response with incomplete platelet recovery (CRi), and 5% partial response (PR)). In both studies, the up-regulation of tumor suppressor genes was observed in responding patients [83].

#### 3.1.3. Panobinostat

Panobinostat (LBH589) is another hydroxamic acid-based HDACi found in all class I, II, and IV HDAC enzymes, with predominant effect on HDAC 1, 2, 3, and 6 and which is implicated in cancer development. This compound has been shown to increase the levels of *CDKN1A* (p21) and to induce hyperacetylation of H3 and H4 [167,168]. By inhibiting the functional activity of all HDACs, this drug promotes the accumulation of the acetylated histones and other non-histone proteins, which induces cell cycle arrest and apoptosis [168,169]. According to several studies, the median IC_50_ values of Panobinostat in different cancer cell lines are significantly lower and had at least ten-fold greater potency when compared with Vorinostat [170]. This drug has been approved by the FDA for the treatment of multiple myeloma (MM) [40,159,160]. In preclinical studies on AML, Panobinostat was shown to modulate the activity of multiple genes, demonstrating a potent anti-leukemic activity in cell lines and primary samples [62,99]. Nevertheless, Panobinostat showed a modest effect as a single agent in early phase clinical trials in advanced hematological malignancies, including AML, showing very low overall and partial response rates [101]. Thus, combination treatment is necessary to achieve a clinical effect. Many drugs have been explored in combination with Panobinostat in AML, such as Decitabine [84,85], AZA [86], BCL-2 inhibitor ABT-199 [87], Wee1 inhibitor MK-1175 [88], β-catenin antagonist BC2059 [89], LSD1 inhibitor SP2509 [90], BRD4 inhibitor JQ1 [91], FLT3 inhibitor AC220 [92], Bortezomib [93], CXCR4 antagonists [94], Doxorubicin [95], or DZnep [96].

All of these combinations showed anti-leukemic effects, inducing apoptosis and inhibiting cell proliferation of AML cell lines as well as primary leukemic cells from patients, and in most cases, improving survival of mice with AML. However, only a few of them have been tested in clinical trials. Preclinically, the combination of Panobinostat with the demethylating agent Decitabine led to a significant reduction in primary AML cell viability compared with either compound alone [85]. Additionally, the use of both compounds led to transcriptional changes in more genes than any drug alone [84]. Similar results were obtained with the combination of Panobinostat and AZA in AML, showing synergistic effects both in vitro and in vivo [86]. However, despite the promising data derived from the first phase Ib/II trial in high risk AML patients [171], the latest results of the clinical trials on AML examining Panobinostat and AZA were to some extent disappointing, with the combination of both drugs not demonstrating a significant advantage in comparison with AZA alone [97]. The combination of Panobinostat with LSD1 inhibitors has also shown synergistic effects [90]. However, a phase I trial with the LSD1 inhibitor GSK2879552 was stopped due to a negative risk–benefit assessment [98]. Finally, the phase Ib/II panobidara study showed that Panobinostat in combination with chemotherapy agents Cytarabine and Idarubicin is a safe and effective treatment for AML patients over 65 years of age. This study evaluated the activity of Panobinostat in combination with chemotherapy, followed by a maintenance phase with Panobinostat in monotherapy [99]. The combination of Panobinostat with Daunorubicin and Cytarabine was well tolerated and the results of this phase I clinical trial were promising [100]. However, another clinical trial showed that the addition of Panobinostat to Cytarabine and Mitoxantrane did not improve the treatment efficacy [101].

#### 3.1.4. Belinostat

Belinostat (Beleodaq or PXD101) belongs to a new class of hydroxymic-type HDACi that acts by blocking zinc-based deacetylase enzymes of classes I, II, and IV (with predominant effect on HDAC1, 2, 3, and 6), thus inducing apoptosis of cancer cells. In several in vitro studies, this drug has shown cytotoxic effects or growth inhibition of solid tumors correlated with hyperacetylation of H4. Belinostat has been approved by the FDA in peripheral T-cell lymphoma [172,173,174]. Regarding AML, Belinostat has demonstrated an effective anti-leukemic effect in AML cell lines, especially in APL cells, promoting cell cycle arrest, inhibiting cell proliferation, and inducing apoptosis. Moreover, in combination with ATRA it accelerates granulocytic differentiation in APL cell lines [102,103]. However, in a phase II clinical study with patients with AML, the effect of Belinostat as a single agent was found to be minimal [175]. Some combinations have been explored in preclinical studies in AML, with the combination of Belinostat with DZnep being the most widely studied. Both compounds in combination with ATRA or ATRA plus Idarubicin induced apoptosis, inhibited cell growth, and enhanced cell differentiation of AML cell lines and patient samples in vitro [104], and increased survival in a mouse model of AML [105]. Additionally, interactions between Belinostat and the proteasome inhibitor Bortezomib, which has been approved for the treatment of refractory MM and mantle cell lymphoma, were investigated in AML. This study showed that Belinostat interacted synergistically with Bortezomib to induce apoptosis in both cultured and primary AML cells [106]. Finally, Zhou et al. reported in their study that the NEDD8-activating enzyme (NAE) inhibitors Pevenedistat and Belinostat interact synergistically by reciprocally disabling the DDR in diverse AML cell types. This Pevenedistat–Belinostat co-administration synergistically induced AML cell apoptosis with or without p53 deficiency or FLT3–internal tandem duplication, supporting further investigations of HDAC/NAE co-inhibitory strategy in AML [107].

Other hydroxamate-based HDACi have been recently tested in clinical studies, such as Givinostat (ITF2357) [176,177,178,179], Resminostat (4SC201) [180], Abexinostat (PCI-24781) [181,182,183,184], Quisinostat (JNJ-26481585) [185], and Pracinostat (SB939) [109,110], with all of them being HDAC pan-inhibitors except for Givinostat, which more specifically targets HDAC1, 2, 3, and 6. However, little information is available regarding AML. Givinostat has potent anti-leukemic activity in vitro and in vivo, inhibiting cell proliferation, inducing apoptosis of AML cell lines and primary samples, and increasing survival in mice with AML [186,187]. Interestingly, those cell lines positive for AML1/ETO were more sensitive to this inhibitor [186]. In the case of Abexinostat, this HDACi was tested in a phase I trial in AML patients with limited clinical benefit [182]. Pracinostat showed potent efficacy on AML in preclinical studies. Furthermore, treatment with Pracinostat in combination with the JAK2 inhibitor SB1518 showed in vitro and in vivo synergism, blocking cell proliferation, inducing apoptosis, and decreasing tumor growth [108]. A phase I clinical trial of Pracinostat alone or in combination with AZA in patients with AML or myelodysplastic syndromes (MDS) demonstrated that the efficacy of this agent was modest as a monotherapy, increasing clinical responses in combination with AZA in MDS patients [109]. Nevertheless, a phase 2 study in older patients with newly diagnosed AML showed very promising results, with CR rate of 42% [110]. Considering these encouraging results, in June 2017 a phase 3 study of Pracinostat in combination with AZA was initiated for the treatment of adults with newly diagnosed AML who are unfit to receive intensive chemotherapy. In this multicenter, double-bind, randomized study, patients will be divided in two groups: an experimental group that will receive Pracinostat plus AZA and a control group that will receive placebo plus AZA. Study treatment is to be continued until disease progression, relapse from complete remission, or unacceptable toxicity. Results for this trial are expected by 2021 [111]. These good results could be related to improved pharmacokinetic and pharmacodynamic properties in contrast to other HDACi, including higher oral bioavailability and accumulation in tumor tissues offering potent efficacy and safety advantages over other HDACi [108,188]. Tefinostat (CHR-2845) is a new pan-HDACi. Tefinostat is cleaved to its active acid by the intracellular esterase human carboxylesterase 1, which is only expressed in monocyte/macrophages and some hepatocytes, resulting in the accumulation of active drug selectively in these cells. This HDACi has been tested in a phase I clinical trial in patients with relapsed or refractory hematological malignancies, showing little clinical effectiveness [189,190]. Finally, Nanatinostat (CHR-3996) is a next-generation hydroxamic acid-based HDACi with greater potency against class I HDACs and promising results in clinical trials [191,192].

### 3.2. Benzamides

Benzamides or amino anilides have been shown to bind to the zinc-chelating moiety for interaction with the catalytic Zn^2+^ in HDAC active sites [167]. These HDACi are selective and potent inhibitors of class I HDACs, but have shown lower activity than hydroxamic acids or cyclic peptides [148].

#### 3.2.1. Entinostat

Entinostat (MS-275) is a synthetic benzamide derivative class I HDACi that potently and selectively leads to cell cycle arrest and hyperacetylation of histones H3 and H4 [167,193,194]. In AML, Entinostat induced growth arrest and apoptosis in cell lines and patient samples, downregulating anti-apoptotic molecules BCL-2 and MCL-1, increasing p21, and inducing acetylation of H3 [195]. The in vivo efficacy of Entinostat has also been demonstrated in a murine AML model [196]. Nishioka et al. reported in their study that Entinostat induced degradation of FLT3 through the inhibition of the chaperon activity of the heat shock protein 90 in AML cells, suggesting that this inhibitor may be useful for the treatment of AML patients with *FLT3* mutations [195]. In a phase I study with AML patients, Entinostat treatment was safe and had cellular and molecular effects, inducing acetylation of H3/H4, expression of p21, and activating caspase 3 in bone marrow mononuclear cells. However, no responses were observed by clinical criteria [197]. Preclinically, synergy of Entinostat with the inhibitor of MEK/ERK AZD6244 [112], the mTOR inhibitor RAD001 [113], and Decitabine [114] has been described. Interestingly, a phase I clinical trial showed that the combination of AZA with Entinostat was effective and tolerable for patients with AML [115]. Nevertheless, a phase II study concluded that the addition of Entinostat to AZA did not increase the clinical response in patients with AML or myelodysplastic syndromes (MDS) [116].

#### 3.2.2. Mocetinostat

Mocetinostat (MGCD0103) is also a selective class I and IV HDACi that has been shown to induce histone hyperacetylation, apoptosis, anti-proliferative activities against a wide range of tumor cell lines, and tumor growth inhibition in multiple cancer models, including hematological malignancies [167]. In AML, Mocetinostat reduced cell viability and induced apoptosis in vitro [198]. Mocetinostat exhibited a safe anti-leukemic activity in a phase I clinical trial in patients with AML or MDS [199]. Subsequent controlled studies, with Mocetinostat alone and in combination with other cytotoxic therapies, will be necessary to characterize the efficacy of this HDACi in AML.

### 3.3. Cyclic Peptides

Cyclic peptides comprise both epoxyketone- and non-epoxy ketone-containing tetrapeptides, and represent the most structurally complicated and diverse class of HDACi [148,200,201,202]. The selectivity of these HDACi against different HDACs depends on their cap group variation. These HDACi can be subdivided into two families: bicyclic depsipeptides (Romidepsin) and cyclic tetrapeptides (Trapoxin A, Apicidin).

#### 3.3.1. Romidepsin

Romidepsin (also known as FK228) is a novel and potent HDACi isolated from *Chromobacterium violaceum*. This natural product is a pro-drug that is activated by cellular reduction to a metabolite-containing a thiol group that chelates the zinc ions in the active center of the HDACs [167]. It is a potent inhibitor of HDAC activity, particularly, it inhibits class I HDAC enzymes better than class II. In addition of histone deacetylation, Romidepsin modulates additional targets involved in cancer initiation and progression such as c-MYC, HSP90, or P53 [201]. In 2009, Romidepsin received the FDA approval for the treatment of the relapsed or refractory cutaneous T-cell lymphoma (CTCL) and peripheral T-cell lymphoma (PTCL) [40,159,160,200]. Romidepsin has in vitro and in vivo effects on AML cells, inducing apoptosis and differentiation of APL cell lines [117,118]. Furthermore, this HDACi induced apoptosis of chemoresistant cells, reversed the gene expression profile of resistant cells, and removed chemoresistant leukemia blasts in a xenograft AML mouse model [203]. Romidepsin was also highly efficient in combination with ATRA [117], Decitabine [118], or AZA [119]. Despite the promising preclinical results, several phase I clinical trials showed that Romidepsin has limited clinical activity in AML patients, at least as a monotherapy [204,205].

#### 3.3.2. Apicidin

Apicidin, a cyclic tetrapeptide, is a natural fugal metabolite that selectively inhibits HDAC2 and HDAC3 in the low nano-molar range and HDAC8 in the high nano-molar range, but does not affect HDAC1 or class II HDACs [206]. It exhibits an anti-proliferative activity against various cancer cell lines and induces selective changes in P21WAF1/Cip1 and gelsolin gene expression, which control cell cycle and cell morphology, respectively. Relatively little is known about the effect of Apicidin on AML. Only one study has demonstrated that this cyclic tetrapeptide induces apoptosis via a mitochondrial/cytochrome-c-dependent pathway, activating caspase-3 in AML cell lines [207].

#### 3.3.3. Trapoxin A

Trapoxin A is a microbial cyclic tetrapeptide that is essentially an irreversible inhibitor of class I HDACs, and it was also found to reversibly inhibit the class II enzyme HDAC6 [208]. Regarding AML, Trapoxin A induced differentiation in combination with ATRA in AML cell lines [120]. Interestingly, Trapoxin A treatment led to the up-regulation of costimulatory and adhesion molecules (CD80, CD86, HLA-DR, HLA-ABC, and ICAM-1) in AML cells, inducing tumor immunity [209].

### 3.4. Aliphatic Acids

Aliphatic acids, the short-chain fatty acids, are structurally the simplest class of HDACi. Due to their weak inhibitory activity, low bioavailability, and fast metabolism, they are the least attractive HDACi [148].

Valproic acid (VPA) is an inhibitor of class I and IIa HDACs that has shown antitumor effects [39,62]. VPA functions as an HDACi, causing in vitro and in vivo hyperacetylation of the N-terminal tails of H3 and H4, most likely by union with the catalytic center, thus blocking substrate access. Furthermore, several preclinical studies have shown that VPA induces differentiation and inhibits proliferation and apoptosis of AML cells [210,211,212,213]. However, there are few clinical studies that have analyzed VPA as a monotherapy in AML, and according to the ones that have, this HDACi has no clinical effect when used as a single-agent therapy for AML [121,137,138].

The effect of the combination of VPA with several drugs in AML has been explored, demonstrating synergy with ATRA [121,122,137,138,139,140], Decitabine [123,124,125], Gemtuzumab ozogamicin (GO) [126], AZA [127], retinoid IIF [128], proteasome inhibitors NPI-0052 or PR-171 [129], Curcumin [130], hydroxiurea or 6-mercaptopurine [131], Dasatinib [132], Bortezomib [133,134], Cytarabine [135,142,143], or the Mouse Double Minute 2 (MDM2) inhibitor Nutlin-3 [136], among others. These combinations induced a synergistic effect on apoptosis and inhibition of cell proliferation of AML cells. However, for most of these combinations there are only preclinical data, and only a few of them have been studied in clinical trials. One of the most widely studied drugs in combination with VPA is ATRA, which has demonstrated anti-leukemic activity in experimental in vitro studies [121,122] but yielded poor responses in several clinical trials in poor-risk or elderly AML patients [137,138,139,140,141]. VPA has been used in combination with Cytarabine, resulting in synergistic anti-leukemic activity in AML cell lines and patient samples and in a murine AML model [135]. In a clinical trial in elderly AML patients, the combination of low doses of Cytarabine with VPA showed good therapeutic activity [142]. However, in a subsequent trial testing this combination in elderly patients with AML, limited clinical activity was shown [143]. The discrepancy between these two trials may be due to the different number of patients, different patient population, or different doses or treatment schedules. Interestingly, a clinical trial demonstrated that the VPA therapy combined with hydroxyurea or 6-mercaptopurin may be effective for advanced AML patients [131]. Finally, the possible benefits of adding VPA to a demethylating agent, such as Decitabine or AZA, have also been studied for AML treatment. The combination of Decitabine with VPA was safe and active, and was associated with transient reversal of aberrant epigenetic marks in a first phase I/II clinical trial in patients with leukemia [125]. Unfortunately, in a second phase I trial testing this combination in patients with AML, the addition of VPA led to encephalopathy, even at low doses [124]. A subsequent phase II study with VPA and low dose of Decitabine did not show any difference in relation to complete response, overall response, or cell survival when comparing the addition of VPA to Decitabine versus Decitabine alone [123]. In the case of AZA, a clinical trial with the triple combination of VPA, AZA, and ATRA was carried out in AML or high-risk MDS patients, demonstrating an overall response of 42% with global DNA demethylation and histone acetylation [127]. Therefore, this triple combination was deemed safe and had significant clinical activity. These results provide significant evidence that VPA is an attractive prospect for use in combination therapy with other anti-leukemic agents for the treatment of AML.

Other HDACi in this aliphatic acid class are B=butyric acid and phenylbutyric acid, which are known to be weak inhibitors of HDAC classes I and II, respectively. Butyric acid enhanced granulocytic maturation of AML cells [214], and just like other HDACi, induced expression of costimulatory and adhesion molecules (CD80, CD86, HLA-DR, HLA-ABC, and ICAM-1) in AML cells, inducing anti-tumor immune responses [209]. A pilot study of phenylbutyric acid in combination with AZA involving patients with AML and MDS showed that this combination strategy is safe and produces biological and clinical outcomes [215].

### 3.5. Electrophilic Ketones

The mechanism of electrophilic ketone HDACi appears to be similar to electrophilic ketone protease inhibitors. The zinc binding in the active site of HDACs is coordinated by the hydrated form of the ketone, which acts in a similar way as a transition state analogue [216]. Several trifluoromethyl ketones and α-ketoamides belong to this group of HDACi. Different studies have shown that trifluoromethyl ketones are active as HDACi, with micromolar inhibitory activity for HDACs, with some of them being submicromolar HDACi with antiproliferative effects [217]. In general, electrophilic ketones showed short half-life because of a rapid reduction in the corresponding inactive alcohols. The fluoride present in some ketones makes these types of ketones, such as trifluoromethyl ketones, much more electrophilic. This is due to the presence of a strong electron withdrawing effect of the fluoride [62]. However, in this case, there is no evidence regarding the anti-leukemic effect of these HDACi.

## 4. Conclusions

Epigenetic therapy is still poorly developed in AML, but it is a very promising field that is rapidly evolving. Different studies have indicated that HDACi have shown some limited effects, such as with single agents in AML. Despite the therapeutic efficacy improvement observed in combination with conventional chemotherapy or other epigenetic inhibitors, the results are still modest. However, some clinical trials with HDACi, especially Pracinostat, in combination with DNA hypomehtylating agents or chemotherapy showed encouraging results. These results place HDACi in a very interesting scenario, in which future studies will be essential to elucidate their potential role as anti-leukemic agents in AML. This is especially important in the case of next-generation HDACi, with which perhaps the long-awaited improvement in the therapeutic response of AML patients might be achieved.

## Figures and Tables

**Figure 1 cancers-11-01794-f001:**
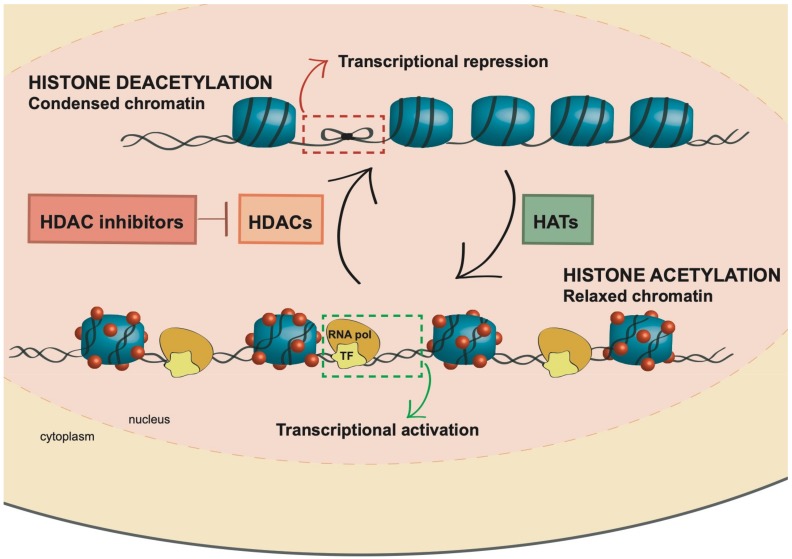
HDAC inhibitor (HDACi) effect on chromatin remodeling. Histone deacetylases (HDACs) and histone acetyltransferases (HATs) are responsible for the balance of histone acetylation, and thereby regulate gene expression. Whereas HDACs deacetylate histones, promoting transcription repression, HATs are responsible for acetylating histones and inducing transcriptional activation. HDACi inhibits HDACs, and thus maintain an open chromatin conformation.

**Table 1 cancers-11-01794-t001:** Description of the HDAC families and their dependence on the Zn^2+^ or NAD^+^ cofactors.

HDAC Classes	HDAC Members	Cofactor
Class I	HDAC1, HDAC2, HDAC3, and HDAC8	Zn^2+^-dependent
Class IIa	HDAC4, HDAC5, HDAC7, and HDAC9	Zn^2+^-dependent
Class IIb	HDAC6 and HDAC10	Zn^2+^-dependent
Class III (Sirtuins)	SIRT1-7	NAD^+^-dependent
Class IV	HDAC11	Zn^2+^-dependent

**Table 2 cancers-11-01794-t002:** Overview of the main HDAC inhibitors and the combinations tested.

HDACi Classes	HDAC Inhibitor	Target HDAC Class	Preclinical Combinations	Clinical Trials Combinations
Hydroximates	Trichostatin A	pan *	Chaetocin [64] Decitabine + DZNep [65]	
	Vorinostat (SAHA)	pan *	ATRA [66,67,68] MK-0457 [69] NPI-0052 [70] Cytarabine [71] Etoposide [71] GX15-070 [72] AZD1775 [73] BPR1J-340 [74] BMN673 [75]	Decitabine [76,77] Idarubicin [78] Idarubicin + Cytarabine [79] Alvocidib [80] GO + AZA [81,82] Sorafenib + Bortezomib [83]
	Panobinostat (LBH589)	pan *	Decitabine [84,85] AZA [86] ABT-199 [87] MK-1775 [88] BC2059 [89] SP2509 [90] JQ1 [91] AC220 [92] Bortezomib [93] CXCR4 antagonist [94] Doxorubicin [95] DZNep [96]	AZA [97] GSK2879552 [98] Cytarabine + Idarubicin [99] Daunorubicin + Cytarabine [100] Cytarabine + Mitoxantrane [101]
	Belinostat (PXD101)	pan *	ATRA [102,103] DZNep + ATRA [104,105] DZNep + ATRA + Idarubicin [104,105] Bortezomib [106] Pevenedistat [107]	
	Givinostat (ITF2357)	pan *		
	Resminostat (4SC201)	pan		
	Abexinostat (PCI-24781)	pan		
	Quisinostat (JNJ-26481585)	pan		
	Pracinostat (SB939)	pan	SB1518 [108]	AZA [109,110,111]
	Tefinostat (CHR-2845)	pan		
	CHR-3996	I		
Benzamides	Entinostat	I	AZD6244 [112] RAD001 [113] Decitabine [114]	AZA [115,116]
	Mocetinostat	I, IV		
Cyclic peptides	Romidepsin	I	ATRA [117] Decitabine [118] AZA [119]	
	Apicidin	I		
	Trapoxin A	I, II	ATRA [120]	
Aliphatic acids	Valproic acid	I, IIa	ATRA [121,122] Decitabine [123,124,125] GO [126] AZA [127] Retinoid IIF [128] NPI-0052 [129] PR-171 [129] Curcumin [130] Hydroxiurea [131] 6-mercaptopurine [131] Dasatinib [132] Bortezomib [133,134] Cytarabine [135] Nutlin-3 [136]	ATRA [137,138,139,140,141] Cytarabine [142,143] Hydroxiurea [131] 6-mercaptopurine [131] Decitabine [123,124,125] AZA [127]
	Butyric acid	I, II		
	Phenylbutyric acid	I, II		

* According to Bradner et al. [144] these HDACi do not demonstrate a preference for Class IIa enzymes at pharmacologically relevant concentrations, suggesting that the target HDAC classes for these HDACi are HDAC I, II, III, and VI. GO: Gemtuzumab ozogamicin; AZA: Azacitidine; ATRA: retinoic acid.

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
