# Peer review of "HDAC Inhibitors in Acute Myeloid Leukemia"

_cancers, 2019, doi:10.3390/cancers11111794_

Round 1

Reviewer 1 Report

This review included general HDAC backgrounds in cancer with focused on AML.  Most HDAC inhibitors do not work in AML as mono-therapy or in combination with other chemotherapeutics drug with exception of Pracinostat that has reached phase III clinical trials.  Although, the authors did mention the pracinostat briefly; however, details are lacking.  Why does pracinostat work but not other HDAC inhibitors?  An extended discussion perhaps will benefit the readers.

In addition, the classification of pan-HDAC inhibitors is a little misleading.  All the so call pan-HDAC inhibitors do not inhibit class IIa well and behave more like HDAC1, 2, 3, and 6 inhibitors (some papers starts to call them that already), but the detailed studies original from Bradner et al Nat. Chem. Biol. 2010 Mar ; 6(3): 238-243.  This probably also need to be adjusted.

Overall, the manuscript is well written.

Author Response

Reviewer 1

Comments and Suggestions for Authors

This review included general HDAC backgrounds in cancer with focused on AML.  Most HDAC inhibitors do not work in AML as mono-therapy or in combination with other chemotherapeutics drug with exception of Pracinostat that has reached phase III clinical trials. Although, the authors did mention the pracinostat briefly; however, details are lacking. Why does pracinostat work but not other HDAC inhibitors? An extended discussion perhaps will benefit the readers.

In addition, the classification of pan-HDAC inhibitors is a little misleading. All the so call pan-HDAC inhibitors do not inhibit class IIa well and behave more like HDAC1, 2, 3, and 6 inhibitors (some papers starts to call them that already), but the detailed studies original from Bradner et al Nat. Chem. Biol. 2010 Mar ; 6(3): 238-243. This probably also need to be adjusted.

We sincerely appreciate the comments of the reviewer. Following his/her suggestions we have included more information about Pracinostat in the revised version of our review manuscript.

Hydroximates section, page 10, lines 352-358: “Considering these encouraging results, in June 2017 a phase 3 study of Pracinostat in combination with AZA was initiated for the treatment of adults with newly diagnosed AML who are unfit to receive intensive chemotherapy. In this multicenter, double-bind and randomized study patients will be divided in two groups: experimental group that will receive Pracinostat plus AZA and control group that will receive placebo plus AZA. Study treatment is to be continued until disease progression, relapse from complete remission or unacceptable toxicity. Results for this trial are expected by 2021 [155].

With respect to why pracinostat works better than other HDAC, the complete mechanism of action of Pracinostat is not yet clearly defined but in vitro studies demonstrate that Pracinostat is a potent oral pan-histone deacetylase inhibitor based on hydroxamic acid. It selectively inhibits HDAC class I, II, IV without class III and HDAC6 in class IV. However, it is well known that Pracinostat has improved pharmacokinetic and pharmacodynamic properties in comparison with other HDAC inhibitors, resulting in high oral bioavailability and accumulation in tumor tissue. Thus, Pracinostat offers potential efficacy and safety advantages over other HDACi. Following the reviewer recommendations, we have included a discussion about this item in the revised version of the review manuscript.

Hydroximates section, page 10, lines 358-361: “These good results could be related to improved pharmacokinetic and pharmacodynamic properties in contrast to other HDACi, including higher oral bioavailability and accumulation in tumor tissues offering potent efficacy and safety advantages over other HDACi [154,156].”

There is some controversy about pan-HDAC inhibitors and there are still plenty current papers describing and calling these HDACi as pan-HDACi (Imai Y, Cancers, 2019; Faiao-Flores T, Clin Cancer Res, 2019; Bocchi L, Int J Mol Sci, 2019; Jeong H, Int J Mol Sci, 2019; Korfei M, PLoS One, 2018; Banerjee NS, Proc Natl Acad Sci, 2018; Eckschlager T, Int J Mol Sci, 2017). Nevertheless, we agree with the reviewer and we have modified it in the Table 2 and throughout the revised version of the manuscript, taking into account the compounds tested by Bradner et al (Apicidin, FK-228, TSA, ITF-2357, PXD-101, Panobinostat, LAQ-824, Vorinostat, Pyroxamide, Tubacin, SHA, APHA, MGCD-0103, MS-275 and CI-994) and subsequently, by Gryder et al. (Vorinostat, TSA, Dacinostat) (Bradner JE, Nat Chem Biol, 2010; Gryder BE, Future Med Chem, 2012).

Table 2. Overview of the main HDAC inhibitors and the combinations tested.

HDACi classes

HDAC inhibitor

Target HDAC class

Preclinical

combinations

Clinical trials

combinations

Hydroximates

Trichostatin A

pan*

Chaetocin [77]

Decitabine + DZNep [78]

Vorinostat (SAHA)

pan*

ATRA [84-86]

MK-0457 [89]

NPI-0052 [90]

Cytarabine [91]

Etoposide [91]

GX15-070 [92]

AZD1775 [93]

BPR1J-340 [94]

BMN673 [96]

Decitabine [98,99]

Idarubicin [100]

Idarubicin + Cytarabine [101]

Alvocidib [102]

GO + AZA [103,104]

Sorafenib + Bortezomib [105]

Panobinostat(LBH589)

pan*

Decitabine [113,114]

AZA [115]

ABT-199 [116]

MK-1775 [117]

BC2059 [118]

SP2509 [119]

JQ1 [120]

AC220 [121]

Bortezomib [122]

CXCR4 antagonist [123]

Doxorubicin [124]

DZNep [125]

AZA [127]

GSK2879552 [128]

Cytarabine + Idarubicin [65]

Daunorubicin+Cytarabine [129]

Cytarabine+Mitoxantrane [112]

Belinostat (PXD101)

pan*

ATRA [133,134]

DZNep+ATRA [136,137]

DZNep+ATRA+Idarubicin [136,137]

Bortezomib [138]

Pevenedistat [139]

Givinostat (ITF2357)

pan*

Resminostat (4SC201)

pan

Abexinostat (PCI-24781)

pan

Quisinostat (JNJ-26481585)

pan

Pracinostat (SB939)

pan

SB1518 [154]

AZA [150,151,155]

Tefinostat (CHR-2845)

pan

CHR-3996

I

Benzamides

Entinostat

I

AZD6244 [166]

RAD001 [167]

Decitabine [168]

AZA [169-170]

Mocetinostat

I, IV

Cyclic peptides

Romidepsin

I

ATRA [176]

Decitabine [177]

AZA [179]

Apicidin

I

Trapoxin A

I, II

ATRA [185]

Aliphatic acids

Valproic acid

I IIa

ATRA [191,194]

Decitabine [197-199]

GO [200]

AZA [201]

Retinoid IIF [202]

NPI-0052 [203]

PR-171 [203]

Curcumin [204]

Hydroxiurea [205]

6-mercaptopurine [205]

Dasatinib [206]

Bortezomib [207,208]

Cytarabine [209]

Nutlin-3 [212]

ATRA [192,193,195,196,213]

Cytarabine [210,211]

Hydroxiurea [205]

6-mercaptopurine [205]

Decitabine [197-199]

AZA [201]

Butyric acid

I, II

Phenylbutyric acid

I, II

*According to Bradner et al. [67] these HDACi do not demostrated a preference for Class IIa enzymes at pharmacologically-relevant concentrations, suggesting that the target HDAC class for these HDACi are HDAC I, II, III and VI.

Accordingly, changes have been including in the revised version of the manuscript:

Hydroximates section, page 7, lines 226-230: Vorinostat (SAHA), was the first marketed HDACi, which promotes protein acetylation, modulates gene expression, induces differentiation, growth arrest, and apoptosis of tumor cells [79–81]. By binding to the active site of HDACs, this drug inhibits class I and class II HDAC enzymes, with predominant effect on class I HDAC, and has shown promising clinical activity against different hematological tumors [79,80].

Hydroximates section, page 8, lines 276-278: “Panobinostat (LBH589) is another hydroxamic acid-based HDACi of all class I, II, and IV HDAC enzymes, with predominant effect on HDAC 1, 2, 3 and 6, implicated in cancer development.”

Hydroximates section, page 9, lines 315-317: Belinostat (Beleodaq or PXD101) belongs to a new class of hydroxymic-type HDACi that acts by blocking Zinc based deacetylase enzymes of classes I, II, and IV (with predominant effect on HDAC 1, 2, 3, and 6), thus inducing apoptosis of cancer cells.”

Hydroximates section, page 9, lines 337-340: “Other hydroxamate-based HDACi have been recently tested in clinical studies, such as Givinostat (ITF2357) [140–143], Resminostat (4SC201) [144], Abexinostat (PCI-24781) [145–148], Quisinostat (JNJ-26481585) [149] or Pracinostat (SB939) [150,151], all of them being HDAC pan-inhibitors, except for Givinostat that targets more specifically to HDAC1, 2, 3, and 6.”

Reviewer 2 Report

The paper is a review of the pre-clinical and clinical data that has been published on HDAC inhibitors in AML. Overall, the paper is comprehensive but needs work. Specifically, please address the following points:

Please review facts stated in introduction paragraph, they are mostly correct but could be more accurate Introduction only discusses standard cytotoxic chemotherapy but does not mention any of the recently approved drugs for AML in the US or internationally over the past few years. Overall, paper reads very dry and needs more character, there is good information but could be presented in a more interesting way. Also, the information about each drug is not presented in a logical format but rather a reporting of all data. It would be better if all the pre-clinical data was presented, followed by the clinical data, followed by the authors interpretation/analysis of the studies and future studies. Recent clinical data is missing, would go back and look up recent studies for each of the drugs It would be helpful to have a table to better visualize all the HDACi and drug combinations that have been studied, as well as current and future combinations Conclusion is overly optimistic and current data is not very promising for its use in AML. Also recommend giving a general summary and discuss future directions. Please review for grammar and punctuation

Author Response

Reviewer 2

Comments and Suggestions for Authors

The paper is a review of the pre-clinical and clinical data that has been published on HDAC inhibitors in AML. Overall, the paper is comprehensive but needs work. Specifically, please address the following points:

Please review facts stated in introduction paragraph, they are mostly correct but could be more accurate Introduction only discusses standard cytotoxic chemotherapy but does not mention any of the recently approved drugs for AML in the US or internationally over the past few years.

We really appreciate the comments of the reviewer and as suggested by him/her we have included the new drugs recently approved for AML in the introduction section of the revised version of our review manuscript.

Introduction section, page 1, lines 41-44: “However, new drugs have been recently approved by the United States Food and Drug Administration (FDA) for AML such as gemtuzumab ozogamicin, enasidenib, midostaurin, glasdegib, venetoclax, ivosidenib or gliteritinib [9].

Overall, paper reads very dry and needs more character, there is good information but could be presented in a more interesting way. Also, the information about each drug is not presented in a logical format but rather a reporting of all data. It would be better if all the pre-clinical data was presented, followed by the clinical data, followed by the authors interpretation/analysis of the studies and future studies.

In this review, we have organized the information of each HDACi presenting first its pre-clinical and clinical data as monotherapy, followed by the pre-clinical and clinical data of the HDACi in combination with other drugs and a finally including a global conclusion and future remarks. We consider that this is another logical way of reporting HDACi information, and change it would mean that the whole manuscript has to be restructured and rewritten. Nevertheless, if the editor and reviewer think that it is necessary, we will be happy to do it.

Recent clinical data is missing, would go back and look up recent studies for each of the drugs

We apologize for not including all recent clinical data and as suggested by the reviewer we have checked all clinical data available for each HDACi and included those missing data in the revised version of our review manuscript..

Hydroximates section, page 8, lines 267-270: “Another phase I clinical trial with Vorinostat in combination with Decitabine and Cytarabine in relapse/refractory AML patients showed that this combination was generally well tolerated with an overall response rate of 35% [107].”

Hydroximates section, page 10, lines 352-358: “Considering these encouraging results, in June 2017 a phase 3 study of Pracinostat in combination with AZA was initiated for the treatment of adults with newly diagnosed AML who are unfit to receive intensive chemotherapy. In this multicenter, double-bind and randomized study patients will be divided in two groups: experimental group that will receive Pracinostat plus AZA and control group that will receive placebo plus AZA. Study treatment is to be continued until disease progression, relapse from complete remission or unacceptable toxicity. Results for this trial are expected by 2021 [155].”

Aliphatic acid section, pages 11, lines 448-450: “One of the most widely studied drugs in combination with VPA is ATRA, which has demonstrated anti-leukemic activity in experimental in vitro studies [191,194] but yielded poor responses in several clinical trials in poor-risk or elderly AML patients [192,193,195,196,213].”

It would be helpful to have a table to better visualize all the HDACi and drug combinations that have been studied, as well as current and future combinations

We agree with the reviewer that a table with all the HDACi and drug combinations would be very useful. Therefore, we have completed the Table 2 including all these pre-clinical and clinical combinations together with their references.

Table 2. Overview of the main HDAC inhibitors and the combinations tested.

HDACi classes

HDAC inhibitor

Target HDAC class

Preclinical

combinations

Clinical trials

combinations

Hydroximates

Trichostatin A

pan*

Chaetocin [77]

Decitabine + DZNep [78]

Vorinostat (SAHA)

pan*

ATRA [84-86]

MK-0457 [89]

NPI-0052 [90]

Cytarabine [91]

Etoposide [91]

GX15-070 [92]

AZD1775 [93]

BPR1J-340 [94]

BMN673 [96]

Decitabine [98,99]

Idarubicin [100]

Idarubicin + Cytarabine [101]

Alvocidib [102]

GO + AZA [103,104]

Sorafenib + Bortezomib [105]

Panobinostat(LBH589)

pan*

Decitabine [113,114]

AZA [115]

ABT-199 [116]

MK-1775 [117]

BC2059 [118]

SP2509 [119]

JQ1 [120]

AC220 [121]

Bortezomib [122]

CXCR4 antagonist [123]

Doxorubicin [124]

DZNep [125]

AZA [127]

GSK2879552 [128]

Cytarabine + Idarubicin [65]

Daunorubicin+Cytarabine [129]

Cytarabine+Mitoxantrane [112]

Belinostat (PXD101)

pan*

ATRA [133,134]

DZNep+ATRA [136,137]

DZNep+ATRA+Idarubicin [136,137]

Bortezomib [138]

Pevenedistat [139]

Givinostat (ITF2357)

pan*

Resminostat (4SC201)

pan

Abexinostat (PCI-24781)

pan

Quisinostat (JNJ-26481585)

pan

Pracinostat (SB939)

pan

SB1518 [154]

AZA [150,151,155]

Tefinostat (CHR-2845)

pan

CHR-3996

I

Benzamides

Entinostat

I

AZD6244 [166]

RAD001 [167]

Decitabine [168]

AZA [169-170]

Mocetinostat

I, IV

Cyclic peptides

Romidepsin

I

ATRA [176]

Decitabine [177]

AZA [179]

Apicidin

I

Trapoxin A

I, II

ATRA [185]

Aliphatic acids

Valproic acid

I IIa

ATRA [191,194]

Decitabine [197-199]

GO [200]

AZA [201]

Retinoid IIF [202]

NPI-0052 [203]

PR-171 [203]

Curcumin [204]

Hydroxiurea [205]

6-mercaptopurine [205]

Dasatinib [206]

Bortezomib [207,208]

Cytarabine [209]

Nutlin-3 [212]

ATRA [192,193,195,196,213]

Cytarabine [210,211]

Hydroxiurea [205]

6-mercaptopurine [205]

Decitabine [197-199]

AZA [201]

Butyric acid

I, II

Phenylbutyric acid

I, II

*According to Bradner et al. [67] these HDACi do not demostrated a preference for Class IIa enzymes at pharmacologically-relevant concentrations, suggesting that the target HDAC class for these HDACi are HDAC I, II, III and VI.

Conclusion is overly optimistic and current data is not very promising for its use in AML. Also recommend giving a general summary and discuss future directions.

We absolutely agree with the reviewer that the HDACi current data are not very good and only there are some encouraging results with some HDACi and always in combination with other drugs. Accordingly, we have modified the abstract and the conclusion as suggested by the reviewer.

Abstract section, page 1, lines 23-27: “In this review we discuss the rationale for the use of different HDACi in patients with AML, the results of preclinical studies and the results obtained in clinical trials. Although so far the results with HDACi in clinical trials in AML are modest, there are some encouraging data with the HDACi Pracinostat in combination with DNA demethylating agents.

Concluding remarks section, pages 12, lines 492-500: “Epigenetic therapy is still poorly developed in AML, but it is a very promising field that is rapidly evolving. Different studies have indicated that HDACi have shown some limited effects as a single agent in AML. Despite the therapeutic efficacy improvement observed in combination with conventional chemotherapy or other epigenetic inhibitors, the results are still modest. However, some clinical trials with HDACi, especially Pracinostat, in combination with DNA hypomehtylating agents or chemotherapy showed encouraging data. These results place HDACi in a very interesting scenario, in which future studies will be essential to elucidate their potential role as anti-leukemic in AML. This is especially important in the case of the next generation HDACi, with which perhaps the long-awaited improvement in the therapeutic response of AML patients might be achieved.”

Please review for grammar and punctuation

Grammar and punctuation have been revised.

Reviewer 3 Report

This is a comprehensive review of HDACi in the treatment of AML. The manuscript is well referenced.

Since these agents have not demonstrated considerable clinical activity in AML, development and experimentation of HDACi for the treatment of AML is fading away. The abstract and conclusion should reflect this fact. Such reflection will not diminish the value of this review article. For example, the last sentence of abstract leaves the impression that combination of HDACi and hypomethylating agents is a novel promising approach, while such combination has been tested extensively over past several years and not yielded clinically significant efficacy over hypomethylating agents alone.

Authours may consider organizing the HDACi in chronological order on the manuscript. For example, valproic acid which was one of the first HDACi tested, may be discussed earlier.

Minor English revisions are preferred, in particular following lines: 16, 46, 62, 108, 148, 191 and 192. In addition, the last sentence of the conclusion is difficult to understand given the length of the sentence. Suggest break it into two sentences.  

Author Response

Reviewer 3

Comments and Suggestions for Authors

This is a comprehensive review of HDACi in the treatment of AML. The manuscript is well referenced.

Since these agents have not demonstrated considerable clinical activity in AML, development and experimentation of HDACi for the treatment of AML is fading away. The abstract and conclusion should reflect this fact. Such reflection will not diminish the value of this review article. For example, the last sentence of abstract leaves the impression that combination of HDACi and hypomethylating agents is a novel promising approach, while such combination has been tested extensively over past several years and not yielded clinically significant efficacy over hypomethylating agents alone.

We sincerely appreciate the comments of the reviewer. We absolutely agree with the reviewer that the HDACi current data are not impressive and only there are some encouraging results with some HDACi and always in combination with other drugs. However, precisely the best results achieved are in combination with a hypomethylating agent, being the case of Pracinostat plus AZA. Accordingly, as suggested by the reviewer, we have modified the abstract and the concluding remarks in the revised version of our review manuscript.

Abstract section, page 1, lines 23-27: “In this review we discuss the rationale for the use of different HDACi in patients with AML, the results of preclinical studies and the results obtained in clinical trials. Although so far the results with HDACi in clinical trials in AML are modest, there are some encouraging data with the HDACi Pracinostat in combination with DNA demethylating agents.

Concluding remarks section, pages 12, lines 492-500: “Epigenetic therapy is still poorly developed in AML, but it is a very promising field that is rapidly evolving. Different studies have indicated that HDACi have shown some limited effects as a single agent in AML. Despite the therapeutic efficacy improvement observed in combination with conventional chemotherapy or other epigenetic inhibitors, the results are still modest. However, some clinical trials with HDACi, especially Pracinostat, in combination with DNA hypomehtylating agents or chemotherapy showed encouraging data. These results place HDACi in a very interesting scenario, in which future studies will be essential to elucidate their potential role as anti-leukemic in AML. This is especially important in the case of the next generation HDACi, with which perhaps the long-awaited improvement in the therapeutic response of AML patients might be achieved.”

Authors may consider organizing the HDACi in chronological order on the manuscript. For example, valproic acid which was one of the first HDACi tested, may be discussed earlier.

We appreciate the suggestion of the reviewer as this is indeed a coherent way to present the results. However, we decided to organize the HDACi according to the group they belong. We consider is a logical way of reporting HDACi information, and change it would mean that the whole manuscript has to be restructured and rewritten. Nevertheless, if the editor and reviewer think that it is necessary, we will be happy to do it.

Minor English revisions are preferred, in particular following lines: 16, 46, 62, 108, 148, 191 and 192.

The sentences indicated by the reviewer have been reviewed in the revised version of our review manuscript.

Abstract section, page 1, lines 15-16: “Although clinical advances in AML have been made, especially in young patients, long-term disease-free survival remains poor, making this disease an unmet therapeutic need.”

Introduction section, page 2, lines 48-49: “Unfortunately, older patients show a high rate of relapse with a notorious poor outcome.”

Introduction section, page 2, lines 63-64: Additionally, we will discuss the novel evidences regarding the synergistic effects, in preclinical and clinical studies in AML, of HDACi with DNA hypomethylating agents and other inhibitors.

HDAC classes section, page 3, line 110: Table 1. Description of the HDAC families and their dependence on the Zn2+ or NAD+ cofactors.

Implication of HDACs in Cancer section, page 4, lines 151-153: Several types of human tumors including gastric, colorectal, liver, breast, lung cancers or hematological malignancies, show aberrant HDACs expression, in many cases associated with advanced disease and poor prognosis in cancer [40–46].

Histone deacetylase inhibitors (HDACi): mechanism of action and role in AML section, page 5, lines 193-196: Furthermore, these inhibitors have shown to induce differentiation, cell-cycle arrest, and apoptosis in AML, leading to a good alternative for treatment, especially for those AML patients not suitable for intensive chemotherapy [23,60].

Histone deacetylase inhibitors (HDACi): mechanism of action and role in AML section, page 5, lines 196-197: Despite the promising preclinical results of HDACi, these HDACi do not seem to be clinically effective as monotherapy in AML.

In addition, the last sentence of the conclusion is difficult to understand given the length of the sentence. Suggest break it into two sentences.

We have modify the Concluding remarks section in the revised version of our review manuscript:

Concluding remarks section, pages 12, lines 492-500: “Epigenetic therapy is still poorly developed in AML, but it is a very promising field that is rapidly evolving. Different studies have indicated that HDACi have shown some limited effects as a single agent in AML. Despite the therapeutic efficacy improvement observed in combination with conventional chemotherapy or other epigenetic inhibitors, the results are still modest. However, some clinical trials with HDACi, especially Pracinostat, in combination with DNA hypomehtylating agents or chemotherapy showed encouraging data. These results place HDACi in a very interesting scenario, in which future studies will be essential to elucidate their potential role as anti-leukemic in AML. This is especially important in the case of the next generation HDACi, with which perhaps the long-awaited improvement in the therapeutic response of AML patients might be achieved.”

Round 2

Reviewer 2 Report

I agree with the revisions.